# Exploratory Analysis of Molecular Subtypes in Early-Stage Osteosarcoma: Identifying Resistance and Optimizing Therapy

**DOI:** 10.3390/cancers17101677

**Published:** 2025-05-16

**Authors:** Luka Bojic, Mina Peric, Jelena Karanovic, Emilija Milosevic, Natasa Kovacevic Grujicic, Milena Milivojevic

**Affiliations:** Institute of Molecular Genetics and Genetic Engineering, University of Belgrade, Vojvode Stepe 444a, 11042 Belgrade, Serbia; luka.bojic@imgge.bg.ac.rs (L.B.); mina.peric@imgge.bg.ac.rs (M.P.); jelena.karanovic@imgge.bg.ac.rs (J.K.); emilija.milosevic@imgge.bg.ac.rs (E.M.); natasa.kovacevic@imgge.bg.ac.rs (N.K.G.)

**Keywords:** transcriptome analysis, osteosarcoma stratification, drug targeting, combined treatment, hesperidin

## Abstract

Osteosarcoma is a malignant bone cancer with little progress in treatment and outcomes for the past 30 years. In this study, we attempted to subtype 102 tumor samples into distinct subtypes based on their gene expression and to explore specific expression profiles that would allow for precise treatment of each subtype. We identified three tumor subtypes with different functional characteristics. One identified subtype was shown to be more resistant to standard chemotherapy drugs and was linked to poorer patient outcomes. We also found that a natural compound, hesperidin, could increase the effectiveness of a typical chemotherapeutic doxorubicin in vitro. These findings could be used to guide more tailored treatment strategies in the early-stages of this disease, ultimately improving the quality of life of these young patients.

## 1. Introduction

Osteosarcoma (OS) is a highly aggressive primary malignant bone tumor that predominantly affects children, adolescents, and young adults, with an incidence of 5 cases per million people under the age of 20 [1]. Although the 5–year survival rate is 65.5% [2], patients with metastases and recurrence have a significantly lower survival rate of ~30% [3]. Despite this concerning statistic the treatment for OS has remained largely unchanged over the past three decades [3]. This stagnation in treatment innovation highlights the urgent need for further research and development of therapies tailored specifically for OS, especially given the aggressive nature of the disease and the significant impact of current treatment regimens on patients’ quality of life [4].

The chemotherapy used for this type of tumor mainly consists of combinations of different drugs, including doxorubicin, methotrexate, cisplatin, ifosfamide, epirubicin, cyclophosphamide, gemcitabine, etoposide, and vincristine [5,6,7]. These drugs are categorized into four distinct classes based on their mechanisms of action: alkylating agents, anti-metabolites, topoisomerase inhibitors, and mitotic inhibitors [8]. However, the high doses required for the treatment of OS patients can have significant drawbacks, leading to short and long-term adverse effects on patients’ health, with hepatotoxicity, nephrotoxicity, neurotoxicity, cardiotoxicity, ototoxicity, hematological, gastrointestinal, and gonadal toxicity being reported [9,10,11,12,13,14,15,16,17,18,19,20,21,22,23,24]. Enhancing therapy and minimizing side effects are crucial particularly for younger patients. Although they are generally more tolerant to higher doses of chemotherapeutics, they are likely to face many more years of living with a reduced quality of life [8]. In recent years, significant attention has been focused on precise and personalized medicine to optimize therapy and enhance quality of life. This approach considers both inter- and intra-tumor variability in gene expression and tumor microenvironment (TME) [25].

The identification of molecular subtypes of OS could lead to more precise treatments tailored for specific subtypes. Currently, classification is mainly based on OS localization and histologic characteristics [26]. On the other hand, molecular classification is still relatively rare and mainly focuses on the TME characteristics or immunological expression profiles [27,28,29,30,31,32]. The challenges associated with characterizing OS present an opportunity for improvement, especially considering the significant toxicity of the chemotherapeutic agents used for treatment. By improving our understanding of OS, we can develop more effective and safer therapeutic strategies.

Bioflavonoids are polyphenolic compounds that are found in various fruits. They are receiving increased attention in cancer management due to their strong antioxidant and bioactive properties [33]. Flavonoids exhibit anti-cancer potential by targeting various molecular signaling pathways. It has been shown that they inhibit cancer growth and differentiation, exhibit anti-inflammatory effects, and contribute to apoptosis induced by antioxidants. They are shown to scavenge reactive oxygen species and induce apoptosis and autophagy in cancer [34]. Furthermore, flavonoids are associated with anti-angiogenic and anti-proliferative properties [34]. Hesperidin, a natural phenolic compound, exhibits anti-cancer properties, as it has been reported to limit the proliferation of various cancer cells in vitro [35].

Despite recent research in the field, prior studies only analyzed OS tumors after neoadjuvant chemotherapy [28] or did not differentiate the patients based on the presence of metastasis [36]. To the best of our knowledge, our study represents the first exploratory research of the processes occurring in the early-stages of OS and before treatment and focuses only on tumors that have not metastasized and have not undergone chemotherapy treatment.

Our study has delineated non-treated, non-metastatic human samples into three molecular OS subtypes which show considerable heterogeneity in functional enrichment, tumor purity and drug sensitivity. We distinguished one OS subtype as having the worst prognosis and identified a prognostic factor for patient survival. Finally, we highlighted hesperidin as a potential co-adjuvant treatment of this most resistant OS subtype.

## 2. Materials and Methods

**Data acquisition:** Publicly available RNA-seq data were downloaded from the Target-OS (https://gdc.cancer.gov/publication-tag/target-os, accessed on 1 March 2025) project and Gene Expression Omnibus (GEO, https://www.ncbi.nlm.nih.gov/geo/, accessed on 1 March 2025) databases. From the Target-OS project data were acquired using the TCGAbiolinks package (version, 2.34.0) [37] in R studio software (version, 4.4.2). From the GEO, we have downloaded the data from GSE87624. The raw data for GSE253548 were provided by Prof. Sulev Koks from the Health Futures Institute, Murdoch University, Australia. Obtained patient datasets were filtered and samples that had not been treated with chemotherapeutics and did not have metastases at the time of sampling were used in the analysis. From the acquired datasets, after filtering and outlier removal, we had 109 samples (102 from OS tumor tissues and seven from healthy bone tissues of OS patients). Batch correction using the Combat_seq algorithm in R [38] was performed on pooled datasets and the obtained data were further used in the analysis (Appendix A). Additionally, datasets GSE39055 and GSE21257 were used as independent cohorts for prognostic factor validation.

From the DepMap database (https://depmap.org/portal, accessed on 1 March 2025) we have downloaded expression data of 19 OS cell lines isolated from primary tumors and used the non-negative matrix factorization (NMF) algorithm to classify cell lines into groups. The obtained differentially expressed genes (DEGs) between the groups were correlated to the DEGs between subtypes. The RNA sequencing data sets GSE197158 and GSE11414 were used to validate aurora kinase B (*AURKB*) and kinesin family member 20 A (*KIF20 A*) expression in OS cell lines SAOS-2 and MG63 compared to normal osteoblast cell lines hFOB1.19 and HOB, respectively.

**Non-negative matrix factorization:** We have utilized the NMF R package (version, 0.28) [39] to identify the molecular subtypes in our OS patient dataset. The input comprised of the 10% of the most variable genes (2741 genes) from 102 tumor samples. Genes with low counts (< 10) were filtered and after variance stabilizing transformation, the variance for each gene was calculated. The nmfEstimateRank function, with 100 iterations was used to determine the cophenetic and silhouette values followed by a non-smooth NMF (nsNMF) algorithm with 500 iterations for the selected cophenetic and silhouette values.

**Differential gene expression analysis and functional enrichment:** DEGs have been identified by DESeq2 R package (version, 1.46.0) [40] by comparison of the gene expression profiles across the identified subtypes (subtype vs. remaining). Adjusted *p*-value < 0.05 and log_2_ Fold Change > 1 were cutoffs for determining DEGs. The identified DEGs were used as input for functional analysis of enriched biological processes with the clusterProfiler (version, 4.14.6) R package [41] and to identify enrichment in the 10 tumor hallmarks (https://cancerhallmarks.com). For visualizing Venn diagrams VennDiagram (version, 1.7.3) R package was used [42]. Further, utilizing DESeq2 we have analyzed changes in gene expression in tumor subtypes compared to normal bone tissue. The obtained upregulated DEGs were used to construct a stringApp (version, 2.2.0) [43] protein-protein interaction (PPI) network in Cytoscape (version, 3.10.1) [44]. The constructed networks were analyzed using MCODE (version, 2.0.3) [45] to identify significant clusters, and CyTargetLinker (version, 4.1.0) [46] was used to extend the obtained networks with drug-target interactions from the DrugBank (https://www.drugbank.com). For the cell line expression data, the limma R package (version, 3.62.2) was used. GSE197158 and GSE11414 were analyzed using the Geo2R website tool to determine DEGs between SAOS-2 and MG63 FOB1.19. and HOB.

**Weighted gene co-expression network analysis:** The weighted gene co-expression network analysis was conducted using the WGCNA (version, 1.73) package in R [47]. Since there was no significant difference in the expression of some genes, we calculated the mean absolute deviation (MAD) for each gene and set the threshold at >0.75 MAD. After filtering, 7297 genes were obtained which were further used to calculate the scale independence and mean connectivity using the pickSoftThreshold function in the WGCNA package (Appendix A). The chosen power of 10, based on the scale of independence and mean connectivity, was further used within the blockwiseModules function for “signed” network construction with a mergeCutHeigth of 0.15 and a minimum module size of 30. Further, we used the biweight midcorrelation (bicor) to calculate the correlation of each module to each identified subtype. The *p*-adjusted values were calculated using the Benjamini-Hochberg method for each module. For visualizing the module-trait heatmap, first, we filtered out the obtained modules by correlation values between −0.3 and 0.3 and then used the labeledHeatmap function to create the module-subtype correlation heatmap. Modules with the highest statistically significant correlation values were used to construct a stringApp PPI network in Cytoscape for visualization and further analysis. After PPI network analysis, we have identified the top ~30 interconnected genes as hub genes. The expression of these genes, with important biological roles, was further characterized by overlapping with upregulated DEGs.

**Assessment of TME score and immune cell infiltration:** Estimation of Stromal and Immune cells in Malignant Tumors using Expression algorithm, ESTIMATE (version, 1.0.13) [48] was utilized to analyze the TME and calculate the stromal score, immune score and the estimate score. The CIBERSORT (version, 0.1.0) [49] algorithm was used to characterize the tumor-infiltrating immune cells (TIICs) of the TME. The fractions of immune cells were quantified using the LM22 as the signature matrix. To determine the significance of the differences between identified subtypes the Kruskal–Wallis test was used with *p*-value < 0.05 considered as significant. Spearman’s correlation was used to calculate the correlations between different immune cell types.

**Drug sensitivity estimation analysis:** OncoPredict (version, 1.2) [50] was utilized to calculate the drug sensitivity of the analyzed pooled dataset. Data from the OncoPredict, precisely the Genomics of drug sensitivity in cancer (GDSC, https://www.cancerrxgene.org/) datasets containing the drug sensitivity and expression information of 805 cell lines and 198 drugs, were used. From this dataset, the data regarding the eight commonly used OS chemotherapeutics were filtered and the estimated IC50 of each identified subtype was calculated. The differences in sensitivity to each chemotherapeutic were analyzed using the Kruskal–Wallis test and *p*-value < 0.05 was considered significant.

**Survival analysis:** Survival analysis was performed in R software, (version, 4.4.2) using survival [51,52], survminer [53], glmnet [54], boot [55,56] and ggplot2 packages [57]. The Univariate Cox proportional-hazards model [58] was used to reveal the association between the survival time of patients (in years) and the expression of selected genes (previously detected as significant for OS subtypes 1–3). The follow-up time for censored data was six months after the last detected death in the specific patient subtype. Median expression value was used as a cutoff to create “high” and “low” gene expression groups. Wald test was used to test the significance of the association. Statistical significance was set at *p*-value < 0.05, while statistical trend (marginal significance) was set at 0.05 < *p*-value < 0.1. The hazard ratio (HR) with a 95% confidence interval (CI) was used as a measure of the strength of an association. To mitigate potential overfitting, penalized regression models (LASSO Cox regression) were employed, and bootstrap resampling improved estimate stability and reproducibility. Additionally, an external validation cohort (datasets GSE39055 and GSE21257) was analyzed to enhance the reliability of the result.

**Cell culture and treatment:** SAOS-2 (ATCC^®^ HTB-85, Manassas, VA, USA) and MG63 (ATCC^®^ CRL1427, Manassas, VA, USA) cells were cultured in Dulbecco’s Modified Eagle Medium high-glucose (DMEM; 4.5 g/L glucose, Gibco, Thermo Fisher Scientific, Horsham, UK) supplemented with 10% fetal bovine serum (PAN Biotech, Aidenbach, Germany) and Penicillin-Streptomycin-Amphotericin B solution (PAN Biotech, Aidenbach, Germany). Cells were maintained at 37 °C in a humidified atmosphere with 10% CO₂. After reaching confluency, cells were passaged and seeded, at a density of 1.5 × 10^4^ for SAOS-2 cells or 0.5 x10^4^ for MG63 cells in 96-well plates. The next day, cells were treated with hesperidin (Sigma-Aldrich, St. Louis, MO, USA) alone or in combination with chemotherapeutics (1 μg/mL, approximating patients’ plasma concentration [59,60]), doxorubicin (Ebewe, Unteracht am Attersee, Austria) or 5-fluorouracil (Sigma-Aldrich, St. Louis, MO, USA) for 24 h, 48 h or 72 h. For the viability assay, the treatments consisted of growing concentrations of hesperidin (20–180 μM) alone or with doxorubicin or 5-fluorouracil.

**MTT Viability assay:** The viability of cells was evaluated using the MTT assay (Sigma-Aldrich, St. Louis, MO, USA) according to the manufacturer’s instructions. Briefly, cells were incubated with MTT solution (0.5 mg/mL in cell culture medium) for 2 h at 37 °C. After incubation, the MTT solution was removed, and formed formazan crystals were dissolved in DMSO (Serva Electrophoresis GmbH, Heidelberg, Germany). Absorbance was measured at 550 nm using a Tecan Infinite 200 PRO microplate reader (Tecan Group Ltd., Mannedorf, Switzerland) and an Epoch plate microplate reader (Agilent Inc., Santa Clara, CA, USA). Data analysis was performed using R studio and One way ANOVA test was used to calculate *p*-values (*p*-value > 0.05, * *p*-value < 0.05, ** *p*-value < 0.01, *** *p*-value < 0.001). Data are presented as a mean value of at least three biological replicates for each time point.

## 3. Results

### 3.1. Identification of Three OS Molecular Subtypes with Distinct Biological and Cancer Hallmark Enrichment Profiles

To classify the 102 obtained non-metastasized, non-treated OS patient samples into subtypes and determine their molecular signatures, the NMF R package was utilized. The optimal rank for the NMF algorithm was determined, based on the calculated cophenetic and silhouette values (Figure 1A) and the analyzed OS samples were classified into three subtypes, S1 (27 samples), S2 (31 samples), and S3 (44 samples) with no difference in survival between them (Figure 1B, Appendix A). Functional enrichment analyses of the DEGs identified for each of the OS subtypes showed that S1 exhibits enriched cell cycle, microtubule-associated processes, and RNA metabolism processes (Figure 1C). On the other hand, subtype S2 had enriched processes related to the extracellular matrix organization, migration, vasculature development and transforming growth factor-beta signaling (Figure 1E). The subtype S3 showed enrichment in cytoplasmic translation, ribosome biogenesis, protein maturation and the Notch signaling pathway (Figure 1G). Further, we utilized CancerHallmarks [61], which provides a consensus gene set associated with ten cancer hallmarks, facilitating a comprehensive enrichment analysis. Using this platform, we identified cancer hallmarks overrepresented in each of the subtypes. All three identified subtypes showed statistically significant enrichment in four hallmarks–sustaining proliferative signaling, resisting cell death, tissue invasion and metastasis, and evading growth suppressors (Figure 1D,F,H). In addition to the identified similarities, our analysis has revealed significant differences between the three subtypes. Namely, S1 has shown a marked enrichment of reprogramming energy metabolism (Figure 1D), while S2 had significant enrichment in the sustained angiogenesis hallmark (Figure 1F). S3 had a statistically significant enrichment in replicative immortality and evading immune destruction (Figure 1H).

Our analysis identified three molecular subtypes of OS that exhibit considerable differences in the enrichment of biological processes. Enrichment analysis of ten cancer hallmarks identified overlapping as well as distinct processes contributing to the pathogenesis of each tumor subtype, emphasizing the heterogeneity of OS.

### 3.2. Gene Co-Expression Networks Reveal Subtype-Specific Functional Enrichment in OS

To identify significantly correlated modules to each subtype, we have utilized the WGCNA package to construct a gene co-expression network for each OS subtype. Our analysis has identified 31 co-expression modules, each presented with their own designated color (Appendix A) and the number of associated genes (Appendix A). For further examination, we have chosen the modules based on the highest correlation values for individual OS subtypes and the top ~30 most interconnected genes (hub genes). Namely, we have identified the pink module as the most significantly correlated to S1 (cor = 0.5; *p*-adj = 6 × 10^−7^ ) while the darkred (cor = 0.49; *p*-adj = 1 × 10^−6^) and grey60 (cor = 0.63; *p*-adj = 3 × 10^−11^) modules were significantly correlated to S2. Finally, the yellow (cor = 0.58; *p*-adj = 2 × 10^−9^) and salmon (cor = 0.65; *p*-adj = 4 × 10^−12^) modules have shown the highest correlation to S3 (Appendix A). To further characterize the identified modules, we analyzed the functional enrichment of Gene Ontology biological processes of each module. The pink module associated with S1 is significantly enriched in processes related to vesicles and intracellular transport (including Golgi, endosomal, and Golgi-to-plasma membrane protein trafficking), p53-mediated signal transduction, protein localization during cytokinesis, circadian rhythm, and microtubule polymerization. From the identified 30 hub genes within the pink module, with the highest degree values, we singled out 12 genes that are significantly upregulated DEGs (Figure 2A) in S1 compared to other OS subtypes. Further, we analyzed the modules that correlated to S2 (Figure 2B). The darkred module is enriched in glycoprotein/oligosaccharide catabolism, endothelial development, platelet-derived growth factor (PDGF) and phosphatidylinositol 3-kinase/protein kinase (PI3 K-AKT) signaling, whereas the grey60 module is predominantly enriched in integrin-mediated cell adhesion, extracellular matrix organization, and chemotaxis. Among the 31 genes in the darkred module, 10 were also found in the upregulated DEGs while in the grey60 module, 21 genes were significantly upregulated in S2 (Figure 2B). After the analysis of the functional enrichment in modules correlated with S3, we have shown that the salmon module is primarily involved in Golgi vesicle transport and ER stress response, whereas the yellow module is enriched in processes related to ossification, osteoblast differentiation, and embryonic skeletal and cardiac development. Out of the 30 hub genes, 14 and 21 genes were upregulated in S3, in the salmon and yellow modules, respectively (Figure 2C).

Overall, from the 31 identified modules we have chosen significant modules with the highest correlation values to individual subtypes. We have shown that the processes enriched in S1 of OS are mainly connected to cellular transport, Golgi apparatus organelle functions and cytokinesis. On the other hand, S2 correlated modules have shown enrichment mostly in extracellular matrix organization, adhesion, PDGF and integrin-mediated signaling. Further, S3 has shown considerable enrichment in Golgi apparatus processes, response to endoplasmic reticulum stress, and bone-forming processes such as ossification and bone development.

### 3.3. TME Analysis Reveals Subtype-Specific Immune Infiltration and Prognostic Implications

In recent years, the microenvironment has been recognized as having a prominent role in tumor development and could consequently provide valuable insights into the pathogenesis of OS [62]. To characterize the identified OS subtypes, we analyzed the TME scores calculated by ESTIMATE. The obtained scores quantify the abundance of stromal and immune cells in tumors, offering insights into TME composition and tumor purity using the ESTIMATE score [63]. We have shown that S1 has a lower immune and stromal score indicating a lower estimate score when compared to other subtypes. S2 has the highest estimate and stromal score when compared to other subtypes (Figure 3A). Further, to better analyze the composition of TIICs we utilized the CIBERSORT algorithm and have shown that S1 had significantly larger fractions of T follicular helper cells, and eosinophils while having the smallest fraction of M2 macrophages when compared to the other identified subtypes (Figure 3B). Also, the first subtype has shown to have larger fractions of T CD8 and B memory cells than S3, while S2 had a larger fraction of resting mast cells and the smallest fraction of dendritic resting cells when compared to the other subtypes. Moreover, the second subtype has shown larger fractions of B memory cells, natural killer (NK) resting cells and activated dendritic cells than S3, while having fewer T gamma delta cells (Figure 3B).

From previous research, it is known that lower ESTIMATE scores (higher tumor purity), are associated with poor prognosis in OS patients [64]. Our study observed significant differences in ESTIMATE scores for the identified OS subtypes, indicating a poor prognosis for S1, which had the lowest score. We have identified significant differences in the fractions of infiltrated immune cells, with M0 and M2 macrophages comprising the largest proportions across all subtypes. In addition, we have shown that S1 has a significantly larger fraction of T follicular helper cells compared to the other two subtypes.

### 3.4. OS Drug Sensitivity

Utilizing the OncoPredict R package, we determined the estimated IC50 values for defined OS subtypes for commonly used chemotherapeutics. In the analysis, we included chemotherapeutics such as methotrexate, etoposide, doxorubicin, gemcitabine, cisplatin, epirubicin, cyclophosphamide, 5-fluorouracil (Figure 4) [5,65,66,67]. Markedly, S1 has shown a statistically significant increase in the resistance to treatment with doxorubicin, cisplatin, epirubicin and 5-fluorouracil when compared to the other subtypes. When compared to S3, S2 has shown significantly higher IC50 values for methotrexate, cisplatin, cyclophosphamide, etoposide, gemcitabine, epirubicin and 5-fluorouracil. On the other hand, S2 is more sensitive to doxorubicin, cisplatin, epirubicin and 5-fluorouracil compared to S1. Finally, we have shown that S3 has the lowest IC50 values among the identified subtypes.

To conclude, S1 has been shown to be the most resistant to the analyzed therapeutics compared to other identified subtypes of OS.

### 3.5. Differential Gene Expression, Drug Targeting and Functional Enrichment

To better characterize the defined subtypes, we identified the DEGs between normal tissues and each subtype. Our analysis revealed that all three subtypes share 384 downregulated genes and 74 upregulated genes. Additionally, we have found that S1 has 368 downregulated and 385 upregulated DEGs, S2 has 161 downregulated and 45 upregulated DEGs, and S3 has 546 downregulated and 203 upregulated DEGs (Appendix A). To conduct a more in-depth study of the obtained upregulated DEGs we have constructed a PPI network and identified the most significant gene clusters. All identified clusters showed enrichment in biological processes connected to cell division and cell cycle positive regulation (Figure 5B, Appendix A).

Furthermore, we extended the networks with information from the DrugBank to identify potential therapeutics for each subtype (Figure 5A, Appendix A). We have focused on analyzing the top 10% of interconnected genes in each cluster due to their important biological functions within each cluster (Figure 5A, Appendix A). Analysis of the S1 subtype, which was shown to exhibit significant resistance to chemotherapy (Figure 4), highlighted hesperidin as a promising candidate for treating this OS subtype. Our analysis revealed that hesperidin targets aurora kinase B (*AURKB*), a gene that has been highlighted in the literature as a crucial factor in OS progression and metastasis [68,69,70]. This IC50 estimation is especially important because subtype S1, which exhibits the highest tumor purity, shows the highest predicted resistance to various commonly used chemotherapeutics compared to other identified subtypes (Figure 5A).

Furthermore, we assessed the association between gene expression and survival probability for the top 10% of interconnected genes. Our analysis has shown that none of the analyzed genes are correlated to survival in S2. On the other hand, *AURKB* marginally increased the hazard for S3 while high expressions of the *PBK* and *CENPA* genes were associated with a reduced hazard for this OS subtype (Appendix A). For S1, high expression of kinesin family member 20A (*KIF20A)* increased the hazard (*p* = 0.02, HR = 21.56, 95% CI = 20.78–22.75), (Figure 5C). Moreover, we have validated our results regarding *KIF20A* using two independent datasets. Using these samples, we have identified *KIF20A* as a gene whose expression significantly impacts survival rates in patients (Figure 5D). It is important to emphasize that, according to the dataset GSE39055 utilized in this validation, 80% of the patients who died displayed elevated expression levels of the analyzed gene. All of these patients had an unfavorable response to chemotherapy with tumor necrosis after chemotherapy being below 90%.

Based on our results, we can conclude that the three identified subtypes share a core group of dysregulated genes, while also having a notable number of DEGs specific to each subtype. The functional enrichment analysis of the three most significant clusters indicates that, despite their differing compositions, they are enriched in similar biological processes related to cell cycle progression and mitosis. Furthermore, our analysis of the survival probability of OS patients revealed and validated its significant association with the expression of the *KIF20A* gene.

### 3.6. The Analysis of Hesperidin’s Effect on OS Cell Lines

To validate our predicted classification of human patient samples into the subtypes, and find adequate cell lines corresponding to each predicted OS subtype, we performed NMF on 19 OS lines obtained from primary tumors within the DepMap portal [71]. Our analysis has shown that all 19 cell lines could be classified into three groups, groups 1, 2, and 3 (Figure 6A). Further, we correlated the significant DEGs (*p*-adj < 0.05) identified between the cell line groups with human sample subtypes. Group 1 strongly correlated with S3 (cor = 0.62). The cell lines in group 3 demonstrate a strong correlation to S2 (cor = 0.89) while group 2 cell lines had the highest correlation to S1 (cor = 0.38) (Figure 6B). To verify drug targeting predictions for OS subtype S1, we selected SAOS-2 and MG63 cell lines which correspond to the S1 subtype. Our results have demonstrated that the expression levels of *AURKB* and *KIF20A*, both of which rank among the top 10% of interconnected upregulated genes in the most significant cluster in S1, are increased in both examined cell lines SAOS-2 and MG63. In SAOS-2, expression levels of *AURKB* and KIF20A have been shown to be increased: by 2.5-fold and 2-fold, respectively, compared to the hFOB1.19 normal osteoblast cell line (Figure 6C), and by 1.25-fold and 1.66-fold, respectively, in MG63 when compared to the HOB normal osteoblast cell line (Figure 6C).

Subsequently, we analyzed hesperidin’s effect on SAOS-2 and MG63 cell viability, both alone and in combination with doxorubicin and 5-fluorouracil, chemotherapeutics to which subtype S1 has shown resistance (Figure 7). Hesperidin has demonstrated a statistically significant effect on the viability of SAOS-2 cells. Hesperidin treatment reduced cell viability by ~30% at concentrations of 100 µM and above, after 24 h, and by 40% at concentrations of 60 µM and above after 48 h. However, we did not observe the effect on cell viability after 72 h of treatment (Figure 7A). In contrast, treatment with 100 µM, 160 µM and 180 µM hesperidin resulted in a significant reduction in the viability of MG63 cells by 30% after 72 h of treatment (Figure 7D).

We have also evaluated the efficacy of hesperidin on both cell lines when combined with doxorubicin and 5-fluorouracil at concentrations approximating those found in patients’ plasma [59,60].

The combined treatment resulted in a further decrease in SAOS-2 cell viability compared to cells treated with doxorubicin alone. Specifically, treatment with doxorubicin for 24 h reduced SAOS-2 cell viability to ~70% while combined treatment with hesperidin at concentrations of 80 μM and above decreased cell viability to around 50% (Figure 7B). Treatment with doxorubicin for 48 h decreased SAOS-2 viability to ~30% while 160 μM and 180 μM hesperidin further reduced the viability to 22% and 18%, respectively (Figure 7B). However, combined treatment with hesperidin in all used concentrations for 72 h did not have any statistically significant effect on SAOS-2 cell viability compared to doxorubicin alone (Figure 7B). This suggests that hesperidin may enhance the anti-cancer effects of doxorubicin, potentially offering a more effective treatment option for targeting SAOS-2 cell viability. The combined treatment with hesperidin did not increase the effect of 5-fluorouracil on the SAOS-2 cell viability (Figure 7C).

After 24 h of combined treatment, 160 and 180 μM hesperidin significantly decreased MG63 cell viability by ~14%, compared to doxorubicin alone (which reduced cell viability to 81.7%) (Figure 7E). Treatment with lower concentrations of hesperidin (20, 60, and 140 μM) for 48 h also resulted in a statistically significant reduction in the MG63 cell viability compared to the chemotherapeutic alone. Furthermore, a 72 h treatment led to an almost complete loss of viability (Figure 7E). Additionally, MG63 cell viability significantly decreased by about 20% after 24 h of treatment with both 5-fluorouracil and 160 or 180 μM hesperidin compared to the chemotherapeutic agent alone (Figure 7F). However, hesperidin did not enhance the cytotoxic effect of 5-fluorouracil on the MG63 cell viability after treatment for 48 h and 72 h (Figure 7F).

Since hesperidin enhanced the effect of doxorubicin on SAOS-2 cells at concentrations of 80, 120, 140, and 180 µM after 24 h of treatment, and at 160–180 µM after 48 h, we analyzed the drug interactions of doxorubicin and hesperidin. For the 24 h treatment, the coefficient of drug interaction (CDI), calculated using the response additivity approach [72], ranged from 0.68 to 1.28 (Appendix A). Although the CDI at 80 µM was 0.68, it was not statistically lower than 1, suggesting an additive rather than a synergistic effect at this concentration. The CDI values at higher concentrations also supported an additive effect between the two compounds after 24 h. However, after 48 h of treatment, the additive effect was no longer observed (CDI > 1) (Appendix A).

In the MG63 cell line, hesperidin enhanced doxorubicin’s effect at 160–180 µM at both 24 h and 48 h. CDI analysis revealed an additive effect at 160 µM and a synergistic effect (CDI < 1) at 180 µM after 24 h, while only additive effects were observed after 48 h (Appendix A). Hesperidin (160–180 µM) similarly enhanced the effect of 5-fluorouracil on MG63 cells after 24 h, with CDI values indicating an additive interaction (Appendix A).

Our findings indicate that the OS cell lines can be categorized into three distinct groups. The correlation of DEGs between these OS cell line groups and the identified OS subtypes has shown a significant relationship, indicating which cell lines might be the best representatives of each identified OS subtype. Consequently, we have selected the SAOS-2 and MG63 cell lines as model systems for analyzing the effects of hesperidin on cell viability. Our analysis revealed that hesperidin significantly increased the effect of doxorubicin, compared to cells treated with doxorubicin alone on both cell lines. On the other hand, hesperidin has increased the effect of 5-fluorouracil only on the MG63 cell line. These results highlight the potential of hesperidin, particularly in combination with doxorubicin, for treating the S1 subtype of OS and underscore the importance of subtype-specific approaches in optimizing therapeutic strategies.

## 4. Discussion

Despite the advances in medicine, the treatment and prognosis for OS have remained unchanged for the last three decades [3]. The current classification, based solely on histology and tumor localization, has proven inadequate, underlining an urgent need for a molecular-based approach that would delineate the different subtypes and enable more precise, targeted treatments. Given that OS primarily arises in people younger than 20 years of age and that conventional therapy has highly toxic effects, we have analyzed the early-stages of disease development before metastasis and chemotherapeutic treatment to shed light on the molecular mechanisms found in each subtype as well as to identify potential therapeutics. This approach would allow for precise treatment of each subtype, potentially lessening the acute and chronic deleterious effects of chemotherapy and improving patients’ quality of life.

Numerous studies have focused on molecular subtyping of OS and have identified subtypes based on gene sets related to various biological processes, cellular senescence, mitochondrial function, the tumor microenvironment, endoplasmic reticulum stress, macrophage activity, Golgi apparatus function, oxidative stress and cancer stem cell characteristics [29,30,73,74,75,76,77,78,79,80,81,82,83]. On the other hand, only a few papers have subcategorized OS samples based on a relatively large number of genes with variable expression across the samples [27,28,36]. This approach facilitates a robust estimate of the subtypes by analyzing multiple aspects and classifying the samples accordingly. A study by Southekal et al. employed multi-omics data, which included gene and miRNA expression, DNA methylation, and associated clinical data of the primary OS tumor samples [27]. This study revealed two distinct subtypes with unique gene expression profiles. Among the differentially expressed genes between the two subtypes were those related to calcium signaling, wingless-related integration site (Wnt), and PI3K-Akt signaling pathways [27] which are linked to the pathogenesis of OS. In our study, the darkred module correlated to S2 is enriched in PI3K-AKT signaling.

Jiang et al. have integrated and analyzed multi-omics data of 121 OS patients after the MAP (methotrexate, doxorubicin, and cisplatin) neoadjuvant chemotherapy and classified OS samples into four consensus subtypes. They singled out one specific subtype as having the poorest prognosis and being the most resistant to chemotherapy, which was attributed to the enrichment in oxidative phosphorylation, a key part of cellular metabolism [28]. Our analysis has shown that high-risk subtype S1 exhibits a significant enrichment in RNA biosynthesis, cell cycle and notably cellular metabolism. The three chemotherapeutics used for treatments of OS patients [29] were also analyzed in our study and we have shown that the S1 subtype is the most resistant to two out of three used drugs. The results of Jiang et al. together with our results reaffirm the significant role of cellular metabolism in OS resistance to chemotherapy. This further highlights the need for precise therapy for each OS subtype to improve the outcome and reduce the adverse effects of chemotherapy.

Zheng et al. classified a dataset of primary OS tumor samples into four molecular subtypes: S-I mainly correlated with bone resorption and osteoclast differentiation; S-II was enriched in immune-related functions; S-III was enriched in processes related to cancer cell proliferation; and S-IV was enriched in lipid metabolism and exhibiting the most unfavorable outcome [36].

Llaneza-Lago et al. developed an unsupervised Bayesian method named latent process decomposition, which considers individual tumor sample heterogeneity for the prediction of OS molecular subtypes [31]. Differential expression analysis and gene onthology enrichment on the subtypes with the poorest prognosis obtained from different datasets delineated a set of eight DEGs shared across all datasets (three upregulated: *ANGPT1*, *CGREF1*, *KAZALD1* and five downregulated: *CILP*, *COL25A1*, *MASP1*, *SDK1*, *SEMA5B*) [31]. Among them, *CGREF1* and *KAZALD1* were found to be coexpressed (*CGREF1*) or upregulated (*KAZALD1*) in a yellow module correlated to subtype S3 in our study which does not exhibit the poorest prognosis.

It is important to note that the previously mentioned studies did not differentiate between the patients who had metastases at the time of sampling and those who did not. Therefore, our study focused on patients prior to chemotherapy, who had not yet developed metastases at the time of sampling. This strategy enables better characterization of these early-stages of pathogenesis and may help determine adequate treatment approaches for each OS subtype. We delineated three subtypes of OS, which exhibit considerable functional heterogeneity in enriched biological processes and tumor hallmarks.

We have successfully identified distinct coexpressed modules significantly correlated to each identified subtype. Each module exhibited a unique set of enriched biological processes. In particular, the module correlated with the S1 subtype had the majority of enriched processes associated with intracellular transport, Golgi apparatus functioning and cytokinesis. It is known that the Golgi apparatus and intracellular transport are important for regulating cell metabolism allowing for increased growth and heightened proliferative capacity of tumors [84]. Notably, many types of tumors, including OS, can sequester and eliminate chemotherapeutics via vesicular transport, extruding them and diminishing the effects of chemotherapy [85,86]. Moreover, Pan et al. have shown that this type of transport and communication between cells in OS can lead to horizontal transfers of resistance to cisplatin treatment [87]. Precisely, they have analyzed the transfer of a circular RNA molecule, via vesicular transport, between cisplatin-resistant and sensitive cell lines. Their result indicates that vesicular transport has a significant role in the spreading of chemotherapy resistance and overall prognosis in OS [87]. Our results indicate that the increased resistance of the S1 subtype could also be explained by this mechanism.

We have analyzed the TME of the identified OS subtypes and revealed significant differences in the TME composition. We have shown that S1 has the lowest ESTIMATE score indicating high tumor purity which is correlated to poor prognosis [64]. We observed that macrophages M0 and M2, make up the largest fractions in all of the subtypes, while the high-risk subtype S1 exhibited a lower M2 fraction compared to the lower-risk subtypes S2 and S3. Moreover, the S1 subtype had significantly larger fractions of follicular helper T cells and eosinophils. The role of the TME immune composition in OS and its correlation to prognosis has been somewhat disputed. The TME composition, and macrophage polarization and infiltration dynamically change in tumors, and the understanding of their role in tumors has to be carefully interpreted considering multiple factors including tumor location, age, metastatic status, and therapy [88]. Deng et al. have shown that higher infiltration of M2 macrophages and follicular helper T cells were correlated to good prognosis [89]. On the other hand, Yang et al. have shown that higher infiltrations of M2 macrophages are correlated with a poor prognosis [90]. Enrichment of the M2 macrophage population has been shown in OS human samples, and upon injection into mice, it has been shown that M2 macrophages contribute to OS initiation and progression [91]. The increased presence of M1 polarized macrophages compared to M2 type was observed in non-metastatic OS [92]. Immunofluorescence analysis of diagnostic biopsies revealed a correlation of high CD163 expression level, which is investigated as one of the markers of M2 macrophages, with longer prolonged non-metastatic period [93]. M2 macrophages expressing CD163 have been found to be correlated with increased vascularization [92] and demonstrated to cause enhanced T cell depletion in OS patients [94]. Understanding the role of macrophages in TME of OS is quite demanding as findings suggest macrophages may act as a double-edged sword, exerting different effects on cancer pathobiology depending on the context [94]. Although the distinction between M1 and M2 polarization is widely used and defines pro- and anti-inflammatory states of macrophages, multiple studies have suggested that macrophages exhibit a multidimensional spectrum of phenotypes in response to various physiological and pathological signals [95], and due to the lack of specificity of marker expression, their distinct between M1 and M2 may simplify the complexity of macrophages, and their role in pathophysiology [96].

Similarly, the role of T follicular helper cells is also highly context-dependent, and it relies on the secretion of different factors, depending on the type of cancer, stage and its unique genetics and phenotypic characteristics thus presenting one of the main challenges in cancer pathobiology [97]. They have shown pleiotropic effects on progression and outcome in different tumors. Namely, these cells produce multiple types of immunologic mediators linked to a better overall immune response to tumors. Gene signatures of T follicular helper cells in tumor tissue are correlated with immune activation and infiltration, tumor burden score, and overall survival in breast cancer and melanoma [97,98,99,100]. On the other hand, the correlation of T follicular helper cells was reviewed by Gutierrez-Melo et al. [97]. Lu et al. have investigated the role of follicular helper cells in OS patient’s peripheral blood. Their results suggest that these cells have significant implications for prognosis with high fractions of these cells in peripheral blood being linked to poor prognosis [101]. In non-small cell lung carcinoma (NSCLC), the existence of different subtypes of T follicular helper cells has been shown. Namely, there has been an increase in T follicular helper cells and a decrease in pro-inflammatory IL-21 in patients’ serum, which indicates impaired function of these cells, possibly causing further immunosuppression and leading to tumor progression in NSCLC [102]. A subtype of T follicular helper-like cells showing inhibitory effect of T cell function has also been shown in human samples and animal models of NSCLC and melanoma [103]. These findings emphasize the diversity of T follicular helper cells cell populations and their different suspected impact on tumor biology [97]. The impact of T follicular helper cell subtypes as prognostic markers in different cancer subtypes has been reviewed in detail elsewhere [97].

When investigating the influence of cancer cells and analyzing the cancer environment it has to be taken into account that the impact of a developing tumor on the host is not limited only to the local TME [88]. The immune microenvironment of osteosarcoma consists of immune cells and non-immune components including mesenchymal stem cells and circulating tumor cells, as well as non-integrated players including complement system and exosomes [104]. Until now, the majority of data we have gathered on cancer biology and physiology have been obtained on human samples collected through biopsy, by removing it from the body and physiologically occurring immune response, or in vitro using cell lines. Excluding the complexity of an immune component in either the biopsied sample or 2D in vitro research limits us in gaining a deeper understanding of the complexity of all players involved in the pathophysiology of cancer. In conclusion, the role of these immune cells in OS TME has yet to be elucidated.

By comparing gene expression profiles in OS subtypes versus normal bone tissue, we have uncovered a shared core of DEGs alongside subtype-specific differences. Within each identified OS subtype, the analysis of the expression profiles revealed a key cluster enriched in processes associated with cell division. This suggests that each OS subtype may utilize distinct mechanisms to regulate cell proliferation, highlighting the importance of targeted therapies that address the specific expression profiles of each distinct subtype. Understanding these unique profiles can aid in developing more effective, personalized therapeutic strategies for OS patients. During the investigation of the top 10% most interconnected genes within the cell division cluster identified in the S1 subtype, *KIF20A* and *AURKB* emerged as particularly significant. *KIF20A* is a member of the kinesin superfamily of microtubule-dependent molecular motor proteins and plays a key role in cell division by transporting chromosomes during mitosis [105]. Furthermore, this gene has been associated with poor overall survival in various tumor types, highlighting its role in tumorigenesis and its correlation to negative outcomes in patients [105]. Zhu et al. have shown that the increase of *KIF20A* expression correlated with poor prognosis in sarcoma patients [106]. Additionally, in a study that analyzed OS progression and metastasis-associated genes, *KIF20A* was recognized as a hub gene in OS, suggesting its important role in the progression of this type of tumor [107]. The upregulation of *KIF20A* expression has also been linked to proliferation in OS U2OS cell lines [108]. Results of our study revealed that *KIF20A* overexpression is associated with poor outcomes and may serve as a new prognostic factor for OS patients. The involvement of *KIF20A* in vesicular transport (15), along with the observed low survival rates associated with its expression, may further substantiate the hypothesis that resistance of S1 subtypes primarily relies on the vesicular sequestration and transport of chemotherapeutics out of the cell.

It is widely recognized that abnormal proliferation of cancer cells is crucial for the development of cancer and its metastasis. In OS, tumor cells are highly proliferative, and display altered metabolic pathways to sustain their uncontrolled growth. To enhance their survival and aggressive growth, cancer cells reprogram their energy metabolism to ensure an adequate supply of energy [109]. We have observed that this cancer hallmark is enriched in the most resistant OS subtype S1 which exhibits overexpression of the AURK family member *AURKB*. *AURKB* plays a vital role in regulating mitosis and is essential for the proper segregation of chromosomes during cell division [110]. The abnormal expression of *AURKB* was associated with pathogenesis and drug resistance in a variety of cancers. *AURKB* affects the proliferation of clear cell renal cell carcinoma by regulating fatty acid metabolism [111]. In OS, *AURKB* is overexpressed and silencing *AURKB* effectively suppresses the malignant phenotype in vitro [70].

The drug targeting analysis conducted in this study highlights hesperidin as a promising therapeutic option for OS, particularly within the most significant cluster in the S1 subtype. Hesperidin, a naturally occurring flavonoid in citrus fruits, is known for its pro-apoptotic effects across various cancers [112]. It also modulates cellular metabolism by reducing glycolysis, commonly referred to as the Warburg effect [112,113]. The effects of hesperidin, well-documented in the literature, suggest its role in cancer prevention and treatment of various types of tumors alone or in combination with different chemotherapeutics [113]. For instance, Shakiba et al. demonstrated hesperidin’s effect on breast cancer cells in vivo, exhibiting significant anti-angiogenic and anti-proliferative [114]. Additionally, hesperidin and hesperetin, have been shown to sensitize hepatocellular carcinoma and lung cancer cells to chemotherapeutic drugs [115,116]. Both hesperetin and neohesperidin decrease cell viability and promote apoptosis and G2 phase arrest of U2OS and HOS OS cell lines [117,118]. Also, hesperetin has been shown to have an additive effect with the chemotherapeutic drug etoposide in reducing OS cell proliferation [117]. In our study, we also found that hesperidin exhibited an additive effect when used alongside doxorubicin. This combined treatment led to a statistically significant reduction in cell viability compared to treatment with either chemotherapeutic alone. These findings suggest that hesperidin could be beneficial as an adjuvant treatment in clinical practice.

It is important to acknowledge several limitations inherent in our study. Firstly, the clinical metadata accompanying the transcriptomic datasets utilized in our analysis were notably incomplete. Additionally, the heterogeneity among studies, particularly regarding the use of various transcriptomic datasets, posed a significant challenge. Key clinical parameters such as survival time, tumor size, and tumor location were either inconsistently reported or absent across different sources. This variability complicates our ability to accurately contextualize the identified molecular subtypes within the broader framework of clinical presentation and undermines our efforts to assess their real-world applicability. Secondly, the sample size presented limitations; while our analysis incorporated transcriptomic data from 102 early-stage osteosarcoma tumors, our control group comprised only seven healthy bone samples. Such a small control cohort may adversely affect the statistical power of our tumor-versus-normal differential expression analyses, potentially skewing the results. Moreover, the correlations drawn between survival outcomes and gene expression were based on a limited follow-up period, which further diminishes the statistical power necessary to relate gene expression profiles to patient prognoses reliably. Lastly, it is essential to note that all assessments of drug efficacy were conducted in vitro. This limitation underscores the pressing need for further research to explore the effects of hesperidin within more complex in vivo environments, which are more representative of actual physiological conditions. Looking ahead, we envision future research directions that will include more comprehensive, systematic, and standardized clinical annotations. Additionally, there is a call to validate our findings across multiple preclinical models and to investigate in vivo efficacy to fully appraise the translational potential of hesperidin as a therapeutic agent for osteosarcoma.

## 5. Conclusions

Our study revealed significant differences in gene expression, functional enrichment, and treatment resistance among the identified subtypes at the early-stages of OS pathogenesis. This highlights the need for improved clinical stratification based on the molecular characteristics of each OS subtype. The presented data emphasize the important role of vesicular transport in OS’s resistance to chemotherapy. We have identified adequate cell models that represent each identified human OS subtype and emphasized the hesperidin’s therapeutic potential in the SAOS-2 and MG63 cell lines, which are representatives of the most resistant subtype, S1. However, further analyses are required to uncover the molecular mechanisms of hesperidin’s effects on OS cells. Understanding the molecular mechanisms will significantly contribute to developing and enhancing existing therapies, thereby facilitating advancements in the treatment of this specific type of tumor. While this study generated important insights, it has several limitations, including the use of incomplete clinical metadata and the heterogeneity of transcriptomic datasets, complicating the interpretation of molecular subtypes. A small control group and a limited follow-up period reduced the statistical power of the findings. Furthermore, all drug efficacy tests were conducted in vitro, emphasizing the need for future in vivo validation. This hypothesis-generating study underscores the importance of stratifying OS patients and offers promising avenues for innovative treatment approaches in OS.

## Figures and Tables

**Figure 1 cancers-17-01677-f001:**
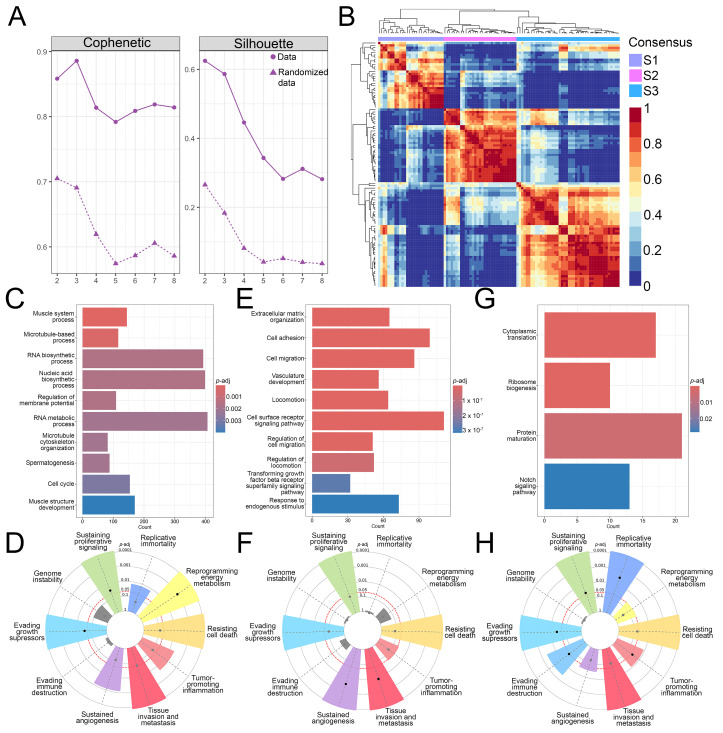
NMF analysis and functional enrichment differences between OS subtypes. (**A**) Cophenetic correlation coefficients and silhouette values were calculated for rank = 2–8. (**B**) NMF heatmap for rank = 3. (**C**–**H**) Gene ontology and tumor hallmarks enrichment of OS tumor subtypes S1 (**C**,**D**), S2 (**E**,**F**), and S3 (**G**,**H**). In the cancer hallmark plots, colored slices represent significantly enriched hallmarks (*p*-adj < 0.05), with larger slices having more significant enrichment. Dots indicate the number of genes within each gene set; gray slices represent non-significant enrichment. The distance of the dots from the central circle is proportional to the number of genes upregulated in the defined gene set. OS–Osteosarcoma; NMF–Non-negative matrix factorization.

**Figure 2 cancers-17-01677-f002:**
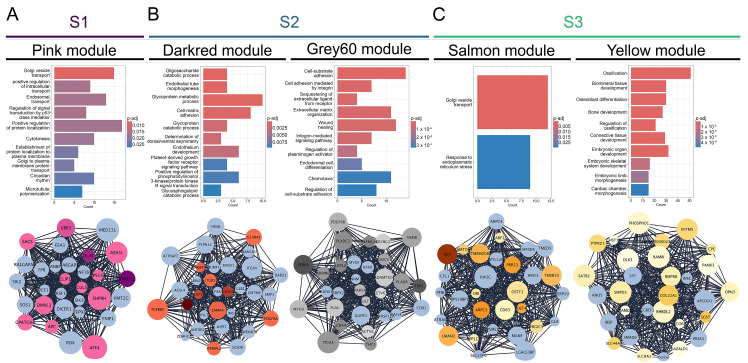
Coexpressed module hub genes and module biological process enrichment. (**A**) S1-correlated pink module (**B**) S2-correlated darkred and grey60 modules (**C**) S3-correlated salmon and yellow modules.

**Figure 3 cancers-17-01677-f003:**
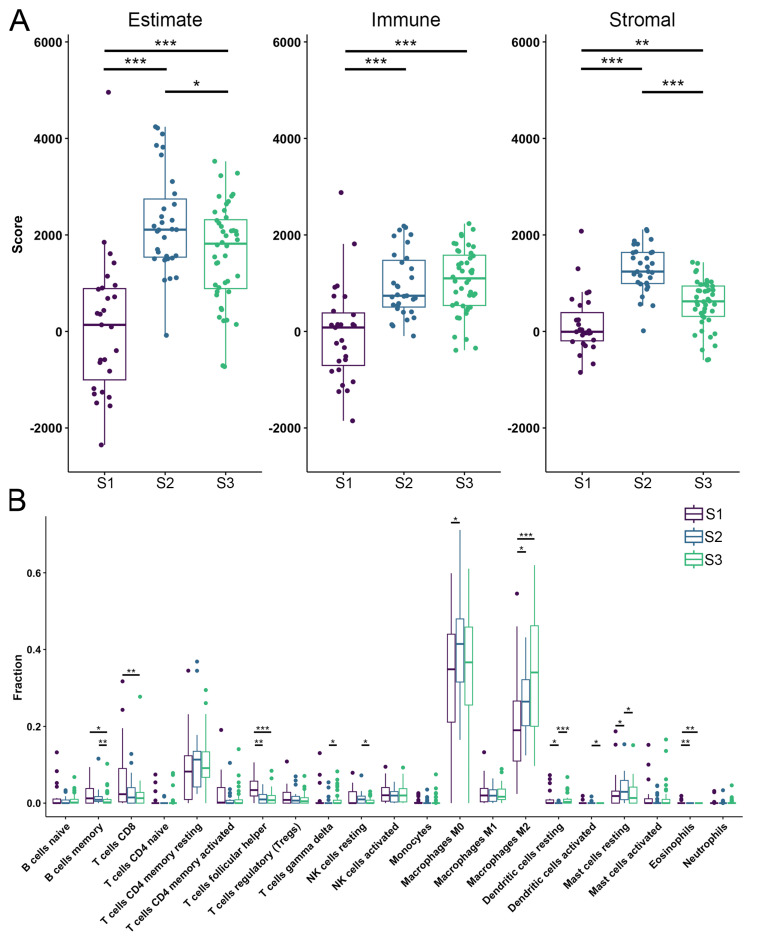
Tumor microenvironment and tumor-infiltrating immune cells. (**A**) Estimate, the immune and stromal scores of each subtype predicted by ESTIMATE algorithm. (**B**) Fractions of TIICs in each identified OS subtype. Purple color—S1; blue color—S2; green color—S3. Statistically significant difference is shown as: * *p*-value < 0.05, ** *p*-value < 0.01, *** *p*-value < 0.001. TIICs–Tumor-infiltrating immune cells; OS–Osteosarcoma.

**Figure 4 cancers-17-01677-f004:**
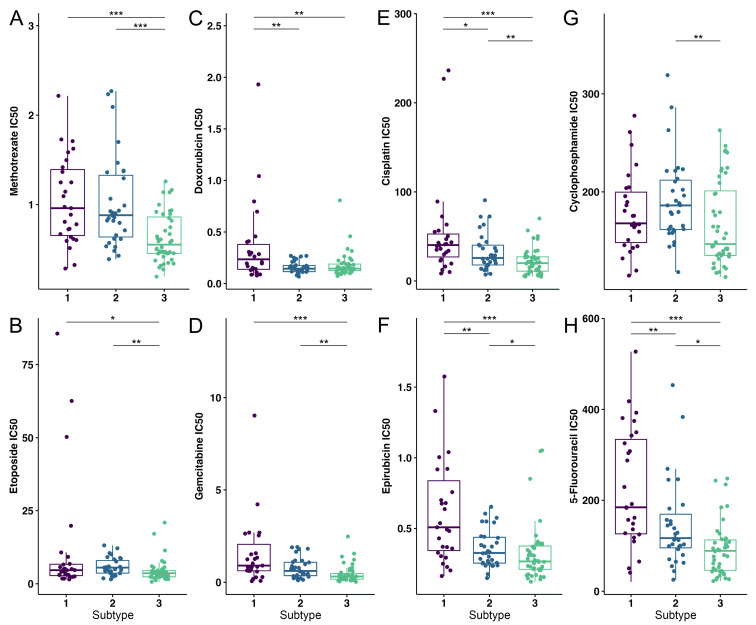
Drug susceptibility analysis of OS subtypes to eight commonly used chemotherapeutics. (**A**) Methotrexate, (**B**) Etoposide, (**C**) Doxorubicin, (**D**) Gemcitabine, (**E**) Cisplatin, (**F**) Epirubicin, (**G**) Cyclophosphamide and (**H**) 5-fluorouracil were the drugs used to test for drug sensitivity. Purple color—S1; blue color—S2; green color—S3. Statistically significant difference is shown as: * *p*-value < 0.05, ** *p*-value < 0.01, *** *p*-value < 0.001. OS–Osteosarcoma.

**Figure 5 cancers-17-01677-f005:**
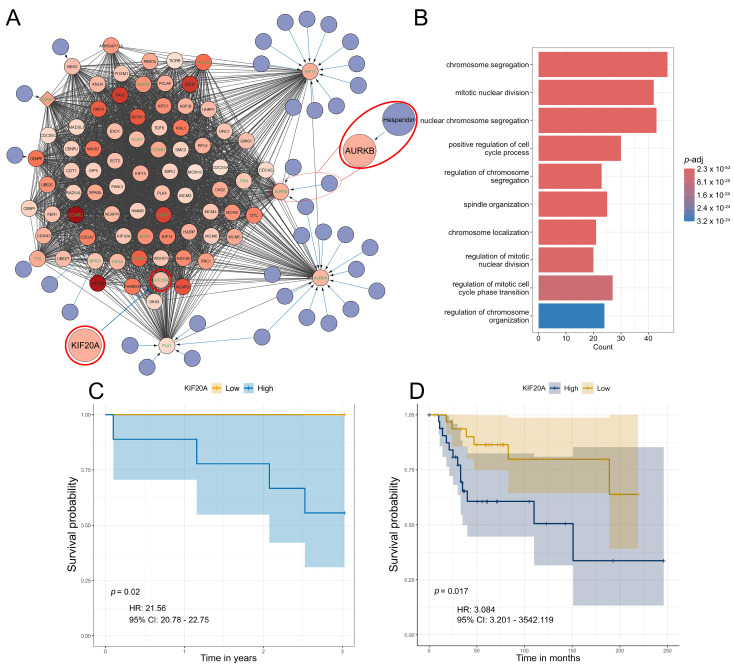
Significant S1 cluster identification and functional enrichment. (**A,B**) Most significant MCODE cluster and the corresponding biological process functional enrichment in S1. (**C**) Survival probability association plot with *KIF20A* in S1. (**D**) Survival probability association plot with *KIF20A* in an independent cohort. Within the identified clusters, genes have been color-coded based on Log2 fold change with increased color density representing higher fold change values. Nodes of the identified drugs are blue and connect to predicted genes via edges. The top 10% based on degree have green gene labels. Hesperidin and its target AURKB are encircled in red. KIF20A is encircled in red.

**Figure 6 cancers-17-01677-f006:**
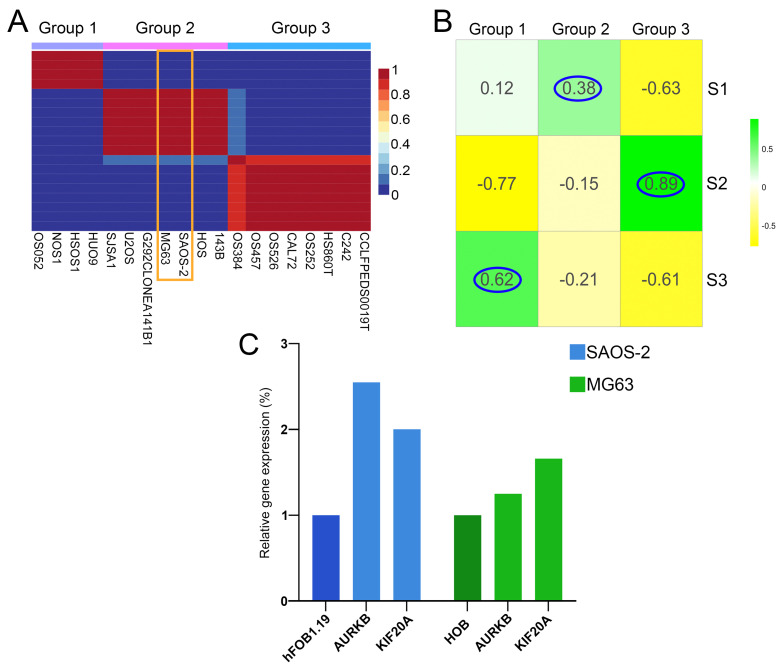
Clustering validation and OS cell line groups correlation to identified patient subtypes. (**A**) NMF cell line heatmap for k = 3, (**B**) Correlation heatmap between cell line groups and patient subtypes. (**C**) Relative expression level of *AURKB* and *KIF20A* genes in SAOS-2 (blue) and MG63 (green) cells compared to hFOB1.19 and HOB. SAOS-2 and MG63 are framed in orange. The highest correlations have been encircled in blue.

**Figure 7 cancers-17-01677-f007:**
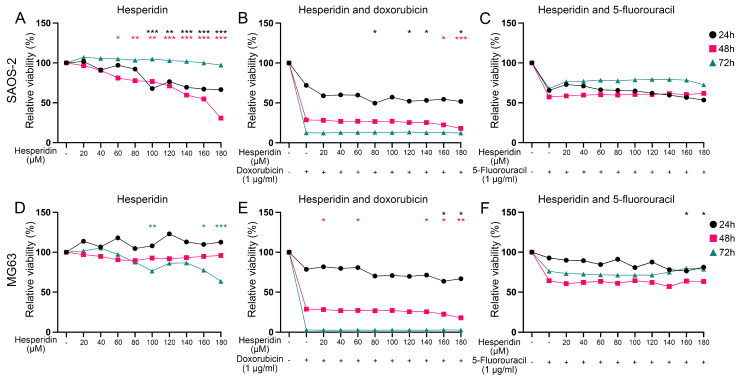
Effect of hesperidin and the combined treatment with chemotherapeutics on OS cell viability after 24 h, 48 h and 72 h of exposure. (**A**,**B**,**C**) Effect of hesperidin alone or in combination with doxorubicin and 5-fluorouracil on SAOS-2 cell viability. (**D**,**E**,**F**) Effect of hesperidin alone or in combination with doxorubicin and 5-fluorouracil on MG63 cell viability. Statistically significant differences are shown as: * *p*-value  <  0.05, ** *p*-value  <  0.01, *** *p*-value  <  0.001. Black circles represent 24 h of exposure, red squares represent 48 h of exposure and green triangles represent values after 72 h of exposure. OS–Osteosarcoma.

## Data Availability

Target-OS (https://gdc.cancer.gov/publication-tag/target-os, accessed on 1 March 2025); GSE87624 (https://www.ncbi.nlm.nih.gov/geo/query/acc.cgi?acc=GSE87624, accessed on 1 March 2025); GSE253548 (https://www.ncbi.nlm.nih.gov/geo/query/acc.cgi?acc=GSE253548, accessed on 1 March 2025); GSE39055 (https://www.ncbi.nlm.nih.gov/geo/query/acc.cgi?acc=GSE39055, accessed on 1 March 2025); GSE21257 (https://www.ncbi.nlm.nih.gov/geo/query/acc.cgi?acc=gse21257, accessed on 1 March 2025); GSE197158 (https://www.ncbi.nlm.nih.gov/geo/query/acc.cgi?acc=GSE197158, accessed on 1 March 2025); GSE11414 (https://www.ncbi.nlm.nih.gov/geo/query/acc.cgi?acc=gse11414, accessed on 1 March 2025); DepMap (https://depmap.org/portal/, accessed on 1 March 2025).

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
