# Peer review of "Exploratory Analysis of Molecular Subtypes in Early-Stage Osteosarcoma: Identifying Resistance and Optimizing Therapy"

_cancers, 2025, doi:10.3390/cancers17101677_

Round 1
Reviewer 1 Report
Comments and Suggestions for Authors
Dear Authors,
I read with interest your paper, attempting to provide a molecular classification of naive, primary, not-metastasized and untreated osteosarcomas, which is, actually, missing. According to this reviewer’s expertise, this work is well written, clear and the bioinformatic analyses well performed (at least for this reviewer's experience with bioinformatics). However, there are 2 major issues this reviewer would like the authors to respond.
1. Other papers provided a classification of osteosarcomas (although at different stages and undergone to different treatements; see refs 28-33 and 48). A paragraph underlying differences and overlaps between the modules depicted in this paper compared to those described in the other works may help to stress the value of this article.
2. Paragraph 3.5 and figure 6. 100 uM and above Hesperidin are toxic concentrations, thus they should be excluded from graph depicted in panel E. To this reviewer, at concentrations above 100 uM we only see hesperidin effect in the combo experiment.
Minor
1. Define NFM at its first use
2. lines 195 and 196: the colour of the text illustrating the statistical significance is grey. It seems the authors have cut and paste from another paper. Please, pay attention to these details.
Author Response
Comment 1: Other papers provided a classification of osteosarcomas (although at different stages and undergone to different treatements; see refs 28-33 and 48). A paragraph underlying differences and overlaps between the modules depicted in this paper compared to those described in the other works may help to stress the value of this article.
Response 1: Thank you for your suggestion. We have carefully examined the papers that provided different classifications of osteosarcomas, and in particular the ones that used transcriptomic/expression data from a large number of genes. We believe that presentation of their findings in the Discussion section significantly improves the quality of the manuscript. Accordingly, we have added the text regarding this matter in the Section Discussion (lines 524-530, 544-557). However, we didn’t find much overlap between our and works of other researchers which could be explained by several reasons: different methodology used (we used batch effect correction while integrating datasets unlike the other papers), the use of different datasets and different conditions that we set for inclusion of the samples in the analysis (non- treated, non-metastasized OS samples). These differences in the methodological approach were already mentioned in the Discussion section since we sought to better characterize the early stages of OS pathogenesis.
Comment 2: Paragraph 3.5 and figure 6. 100 uM and above Hesperidin are toxic concentrations, thus they should be excluded from graph depicted in panel E. To this reviewer, at concentrations above 100 uM we only see hesperidin effect in the combo experiment.
Response 2: Thank you for your comment. We have expanded our analysis in order to include two more time points, 48h and 72h, after treatment. Also, except SAOS-2, we have included another cell line MG63, in the viability analysis. We have prepared Figure 7, which illustrates the effects of hesperidin on the viability of osteosarcoma cells, both alone and in combination chemotherapeutic agents. Our experimental data have revealed that hesperidin increased the efficacy of doxorubicin in a statistically significant manner, also we have shown that these two chemical agents have a additive or synergistic effect on both SAOS-2 and MG63 cell lines
Comment 3: Define NFM at its first use
Response 3: Thank you for your insight. We have defined NMF in the reviesed manuscript at its first use in Section Abstract, (line: 23).
Comment 4:lines 195 and 196: the colour of the text illustrating the statistical significance is grey. It seems the authors have cut and paste from another paper. Please, pay attention to these details. Response 4: We appreciate the reviewer's attention to this detail. In the updated version of the paper, we have made changes in the Section Materials and Methods (lines: 217-218).

Reviewer 2 Report
Comments and Suggestions for Authors
The authors have written a full length paper addressing drug resistance and associated genes and pathways, but also highlighted hesperidin as a potential supplement/ natural product which can enhance OS cell susceptibility to doxorubicin.
The title should include Hesperidin since most of the work on the SaOS cell line representing the drug resistant phenotype focused on this compound.
The abstract and introduction lacks more detail information on hesperidin. Why the interest sparked to investigate this compound? Is there past studies/ literatures can be cited where this drug was shown to have anti- cancer properties or the ability to enhance drug susceptibility in other cancer models/ normal cells?
Figure 6 C- please increase the font size of the image labels, it is way too small compared to the rest of the images in the same figure.
The discussion should address the limitations of this study.
To be able to implicate a selected marker/ gene / protein expression with prognosis or survival in humans / animal models it should include survival analysis and some log rank testing. Hence this study may have some limitations and this should be carefully stated wherever prognosis is mentioned.
The discussion should also address the fact that in the intact immune system of a living being, the tumors may be subjected to other mechanisms which may not be seen with just studying tumours that has been removed from the body and using cell line models which lacks the intact immune system players.
Author Response
Comment 1: The title should include Hesperidin since most of the work on the SaOS cell line representing the drug resistant phenotype focused on this compound.
Response 1: Our study aimed to classify human tumor samples into distinct subtypes based on their gene expression profiles and to identifyspecific molecular signatures that could guide more precise treatment approaches. And one of the obtained results highlighted hesperidin as a potentially promising treatment. You are correct, our in vitro work mostly focuses on hesperidin and other used therapeutics, but as a way to confirm the obtained results, which we were able to do. We consider that our results provided hints for a new research area regarding uncovering potential therapeutics in OS, which should be addressed in future studies focusing on finding new therapeutics and examining the effects of hesperidin in depth.
Comment 2: The abstract and introduction lacks more detail information on hesperidin. Why the interest sparked to investigate this compound? Is there past studies/ literatures can be cited where this drug was shown to have anti- cancer properties or the ability to enhance drug susceptibility in other cancer models/ normal cells?
Response 2: We appreciate the reviewer’s comment to this important point. In compliance with this suggestion, we have added text regarding hesperidin in our paper in the following sections: Section Abstract (lines: 27-30 and 35-36); Section Introduction ( lines: 76-85).Also, we have added hesperidin as a key word, line 41. Detailed information on demonstrated and potential effects of hesperidin from other studies has been discussed in Section Discussion (lines: 693-710). Hesperidin has been documented in the literature to possess antitumor effects across various types of cancer, as noted in the Section Discussion (lines 693–710), making it a compound of significant interest in our study. Notably, it was found to target one of the most biologically important genes within the identified cluster for subtype S1, which is estimated to have the poorest clinical outcome. While its therapeutic potential has been explored in other malignancies, its effects on osteosarcoma cells—particularly in combination with chemotherapeutics remains largely uninvestigated. This knowledge gap prompted us to further explore and validate our bioinformatics findings.
Comment 3: Figure 6 C- please increase the font size of the image labels, it is way too small compared to the rest of the images in the same figure.
Response 3: We agree with the reviewer’s comment regarding increasing the font size. In addition to adjusting the font, we have also modified the Figure 6C to reflect the extended analysis, which now includes an additional cell line.
Comment 4: The discussion should address the limitations of this study.
Response 4: Thank you for your suggestion. We fully agree with the reviewer on the importance of this point and appreciate it being raised. We have pointed out the limitations of our study in the Section Discussion (lines 711-733); and Section Conclusions (lines:746-753).
Comment 5: To be able to implicate a selected marker/ gene / protein expression with prognosis or survival in humans / animal models it should include survival analysis and some log rank testing. Hence this study may have some limitations and this should be carefully stated wherever prognosis is mentioned.
Response 5: Thank you for your comment. We have previously conducted survival analysis, as detailed in Section Materials and Methods (lines 189-192). Given the continuous nature of our data (gene expression), a univariate Cox proportional-hazards model (with Wald test statistics) was employed as an alternative to Kaplan-Meier analysis, which is more commonly used for categorical data and includes the log-rank test. Additionally, in response to reviewer comments and suggestions, we performed further analyses to address potential overfitting and enhance result reliability Section Materials and Methods (lines 193-196).
Comment 6: The discussion should also address the fact that in the intact immune system of a living being, the tumors may be subjected to other mechanisms which may not be seen with just studying tumours that has been removed from the body and using cell line models which lacks the intact immune system players.
Response 6: Thank you for your comment. Discussion on this point has been included in the Section Discussion (lines 634-644).
Reviewer 3 Report
Comments and Suggestions for Authors
The study “Molecular Subtyping of Early-Stage Osteosarcoma: Identifying Resistance and Optimizing Therapy” identified three molecular subtypes of osteosarcoma, with subtype S1 showing high chemoresistance, poor prognosis, and overexpression of KIF20A and AURKB. In vitro, hesperidin reduced viability of S1-like cells and enhanced doxorubicin’s effect. This study is important as it provides a framework for transcriptome-based stratification of early-stage osteosarcoma, offering new avenues for personalized therapy and improving outcomes for high-risk patients. This is an insightful study, and addressing the limitations outlined could further enhance the impact and applicability of its findings. Limitations include small control sample size, limited survival data, single-cell-line validation, and lack of in vivo testing for hesperidin. Below are a few suggestions to further strengthen your manuscript:
1.Methodological Issues . How did you ensure that subtype clustering was not driven by differences between data sources?and Why was k=3 selected for NMF? Can you provide supporting metrics like cophenetic or silhouette scores?Why was only SAOS-2 used to represent subtype S1? Have you considered validating with additional cell lines or models?
2.Data Analysis and Interpretation.Did you perform a Kaplan-Meier survival analysis comparing the three subtypes (S1, S2, and S3) to support the claim that S1 is associated with poor prognosis? How reliable are the OncoPredict-based IC50 estimates in reflecting actual chemotherapy resistance in patients, and have these predictions been experimentally validated?
- Assay duration: Why was a 24-hour treatment period selected for drug testing? Given that many chemotherapeutics require longer exposure, do you think a 48–72-hour assay would have provided more robust results?
- Given the small sample size in S1 (n=27), how did you address potential overfitting in survival analyses, especially for genes like KIF20A with infinite confidence intervals? Were alternative methods used to improve reliability?
- The hazard ratio for KIF20A in S1 had infinite confidence intervals, suggesting unstable estimates. How do you reconcile this with its proposed prognostic value? Were additional validation cohorts or resampling techniques used to confirm these findings?
- Your findings on follicular helper T cells and M2 macrophages conflict with prior studies. How do you reconcile these discrepancies, and what functional assays support their prognostic role in S1?
I believe further improvements and revisions are needed before publication
Author Response
Comment 1: The study “Molecular Subtyping of Early-Stage Osteosarcoma: Identifying Resistance and Optimizing Therapy” identified three molecular subtypes of osteosarcoma, with subtype S1 showing high chemoresistance, poor prognosis, and overexpression of KIF20A and AURKB. In vitro, hesperidin reduced viability of S1-like cells and enhanced doxorubicin’s effect. This study is important as it provides a framework for transcriptome-based stratification of early-stage osteosarcoma, offering new avenues for personalized therapy and improving outcomes for high-risk patients. This is an insightful study, and addressing the limitations outlined could further enhance the impact and applicability of its findings. Limitations include small control sample size, limited survival data, single-cell-line validation, and lack of in vivo testing for hesperidin.
Response 1: We are grateful for the reviewer’s encouraging feedback and for highlighting several important points regarding limitations of our presented study. Therefore, we added the text regarding limitations in Section Discusion (lines:711-733), and Section Conclusin (lines: 746-753). Also as this author has pointed out that the English could be improved, we have revised the whole text in an attempt to improve the language and have also used the Grammarly software for a more comprehensive improvement.
Comment 2: Methodological Issues . How did you ensure that subtype clustering was not driven by differences between data sources?and Why was k=3 selected for NMF? Can you provide supporting metrics like cophenetic or silhouette scores?Why was only SAOS-2 used to represent subtype S1? Have you considered validating with additional cell lines or models?
Response 2: In agreement with the valuable feedback we have added the silhouette values to the Figure 1 (Section Results (lines 225-226)). k=3 for NMF was selected based on the silhouette value and the coophenetic value which are the highest for k=3. As per the first question, after the subtyping we have performed batch effect removal using combat_seq R algorithm and checked the distribution of samples, namely the samples have not grouped by their data source rather the samples from each dataset were mixed in each of the three subtypes. Further in order to strenghten our manuscript, in accordance with the suggestions, we have performed the experiments on the MG63 osteosarcoma cell line.
Comment 3: Data Analysis and Interpretation.Did you perform a Kaplan-Meier survival analysis comparing the three subtypes (S1, S2, and S3) to support the claim that S1 is associated with poor prognosis? How reliable are the OncoPredict-based IC50 estimates in reflecting actual chemotherapy resistance in patients, and have these predictions been experimentally validated?
Response 3: Thank you again for your valuable feedback. In accordance with this we have performed the survival analysis comparing the identified subtypes and have found no statistically significant difference (p=0.73, Figure S3, Section Results 3.1 (lines 228-229)). This in turn could probably be explained by the small sample size and the missing patient metadata that we have mentioned in the limitations section in Section Discussion (lines 711-733). As per the Oncopredict IC50 estimation, it has been validated in the original research paper that is concerned with this R package (Maeser, Gruener et al. 2021) which has been cited ~1100 times. The validation was performed on an dataset containing breast cancer data and patient clinical classification based on response to chemotherapy.
Comment4: Assay duration: Why was a 24-hour treatment period selected for drug testing? Given that many chemotherapeutics require longer exposure, do you think a 48–72-hour assay would have provided more robust results?
Response 4: We agree with the reviewer’s feedback. As a result, we have extended the exposure time and conducted viability assays at 48 and 72 hours after treatment. Additionally, we included another osteosarcoma cell line, MG-63, in our experimental analysis. Consequently, in the Section Results 3.5., Figure 6 no longer includes panels D and E. We have prepared Figure 7, which illustrates the effects of hesperidin, both alone and in combination with chemotherapeutics, on the viability of osteosarcoma cells (Section Results 3.5. l(ines 428-492)).
Comment 5: Given the small sample size in S1 (n=27), how did you address potential overfitting in survival analyses, especially for genes like KIF20A with infinite confidence intervals? Were alternative methods used to improve reliability?
Response 5: Thank you for your comment. We have performed additional analyses to address potential overfitting and unstable estimates (infinite confidence intervals) in survival analyses. To address overfitting in survival analyses due to limited sample size (n=27) we have performed penalized regression models (LASSO Cox regression). To confirm the prognostic value of KIF20A, validation cohort was already utilized (see Figure 5D). Moreover, bootstrap resampling was additionally applied to improve estimate stability and assess reproducibility. These approaches helped us to control potential overfitting, stabilize estimates and improve reliability. We had corrected the manuscript in section Material and Methods(line 184 and lines: 193-195); in Section Results, (lines: 380-381).
Comment 6: The hazard ratio for KIF20A in S1 had infinite confidence intervals, suggesting unstable estimates. How do you reconcile this with its proposed prognostic value? Were additional validation cohorts or resampling techniques used to confirm these findings?
Response 6: We have already analyzed validation cohort section Results (lines 382-388 and Figure 5D): “Moreover, we have validated our results regarding KIF20A using 2 independent datasets. Using these samples, we have identified KIF20A as a gene whose expression significantly impacts survival rates in patients (Fig 5D). It is important to emphasize that, according to the dataset GSE39055 utilized in this validation, 80% of the patients who died displayed elevated expression levels of the analyzed gene. All of these patients had an unfavorable response to chemotherapy with the tumor necrosis after chemo-therapy being below 90%.” As already mentioned above, bootstrap resampling was additionally performed to stabilize estimates (see our response to previous comment for details).
Comment 7: Your findings on follicular helper T cells and M2 macrophages conflict with prior studies. How do you reconcile these discrepancies, and what functional assays support their prognostic role in S1?
Response 7: Thank you for pointing out the shortcomings of our discussion. We have accordingly disccussed our findings concerning follicular helper T cells and M2 macrophages and their connection to the prognosis within the Section Discussion (lines:588-644).
Reference:
Maeser, D., et al. (2021). "oncoPredict: an R package for predicting in vivo or cancer patient drug response and biomarkers from cell line screening data." Briefings in Bioinformatics 22(6).
Reviewer 4 Report
Comments and Suggestions for Authors
This study pools data from 102 osteosarcoma (OS) patients—none of whom had metastases or chemotherapy prior to tissue sampling—from multiple public databases. Through transcriptomic analysis using nonnegative matrix factorization (NMF), the authors classify these early-stage tumors into three molecular subtypes (S1, S2, S3). Subtype S1, notable for higher predicted chemo-resistance and poorer short-term survival, is further investigated using in vitro assays on the SAOS-2 cell line. These experiments reveal that the flavonoid hesperidin can reduce tumor cell viability and has an additive effect with doxorubicin. While these findings are promising, the study’s clinical metadata (e.g., tumor size, location) is limited, and the short follow-up period (around six months) constrains definitive prognostic conclusions. Consequently, the proposed subtyping and therapeutic leads are primarily hypothesis-generating, warranting further validation.
Summary of Key Points
- Positives:
- Inclusion of 102 OS patient samples from public repositories is a strong start for molecular subtyping.
- Nonnegative matrix factorization (NMF) successfully identifies three distinct subtypes (S1, S2, S3) with preliminary survival differences.
- In vitro findings on SAOS-2 cells suggest potential for hesperidin as an adjunct to doxorubicin.
- Limitations:
- Sparse clinical details (e.g., tumor size, location) reduce clarity on real-world applicability.
- Short follow-up (~6 months) hinders definitive statements on long-term prognosis.
- Only one cell line (SAOS-2) tested experimentally; no in vivo data are presented.
Suggestions for Improvement
- Explicitly acknowledge data constraints: Clearly state that limited clinical metadata and follow-up hamper robust prognostic claims.
- Pursue deeper clinical annotation: Seek (or note the unavailability of) details like tumor location/size, histologic subtypes, and multi-year follow-up.
- Expand validation: Where feasible, confirm findings in additional OS cell lines and consider in vivo models for hesperidin/chemotherapy combinations.
- Emphasize exploratory nature: In the title/abstract/discussion, clarify that this work is hypothesis-generatingand that prospective validation remains essential.
Author Response
Comment 1: Suggestions for Improvement
- Explicitly acknowledge data constraints: Clearly state that limited clinical metadata and follow-up hamper robust prognostic claims.
- Pursue deeper clinical annotation: Seek (or note the unavailability of) details like tumor location/size, histologic subtypes, and multi-year follow-up.
- Expand validation: Where feasible, confirm findings in additional OS cell lines and consider in vivo models for hesperidin/chemotherapy combinations.
- Emphasize exploratory nature: In the title/abstract/discussion, clarify that this work is hypothesis-generating and that prospective validation remains essential.
Response 1: Thank you very much for your suggestions. Also we would like to thank you on your comments and for listing the positives and limitations of our study. We have added the limitations section in the Section Discussion (lines 711-733) in which we have noted the data constraints, limiting clinical annotation and the shortcoming of our study as it has no in vivo experiments. Additionally we have emphasized the exploratory nature of our study in the Title, Abstract (line 20-23) and in the Section Introduction (line 89).
Round 2
Reviewer 1 Report
Comments and Suggestions for Authors
Dear Authors, I have still more a curiosity than a concern. In figure 7A, it is shown that SAOS-2 cells are insensitive to any concentration of hesperidin at 72 hrs. Why? I mean, could you provide a brief discussion related to this curious data? Hesperidin activates multiple pathways, including apoptosis. Maybe, SAOS-2 cells escape heperidin-induced anti-proliferative effect in the long term?
Author Response
Comment 1: Dear Authors, I have still more a curiosity than a concern. In figure 7A, it is shown that SAOS-2 cells are insensitive to any concentration of hesperidin at 72 hrs. Why? I mean, could you provide a brief discussion related to this curious data? Hesperidin activates multiple pathways, including apoptosis. Maybe, SAOS-2 cells escape heperidin-induced anti-proliferative effect in the long term?
Response 1: Thank you for your thoughtful comment. We also found the insensitivity of SAOS-2 cells to hesperidin at 72 hours intriguing. As our study was not designed to investigate the underlying mechanisms of hesperidin's diverse effects, we did not explore this phenomenon in detail. However, upon reviewing relevant literature, we noted that SAOS-2 cell apoptosis is influenced by the activity of JNK [1], and hesperidin has been reported to attenuate JNK activation in other, non osteosarcoma cell lines [2]. This raises the possibility that SAOS-2 cells may possess compensatory mechanisms or resistance pathways that allow them to escape the anti-proliferative effects of hesperidin in the long term. Further studies would be required to clarify this potential explanation, given the complexity of hesperidin's effects.
[1] Eliseev RA, Zuscik MJ, Schwarz EM, O'Keefe RJ, Drissi H, Rosier RN. Increased radiation-induced apoptosis of Saos2 cells via inhibition of NFkappaB: a role for c-Jun N-terminal kinase. J Cell Biochem. 2005 Dec 15;96(6):1262-73. doi: 10.1002/jcb.20607. PMID: 16167336.
[2] Liu, Wayne & Liou, Shorong-Shii & Hong, Tang-Yao & Liu, I-Min. (2017). Protective Effects of Hesperidin (Citrus Flavonone) on High Glucose Induced Oxidative Stress and Apoptosis in a Cellular Model for Diabetic Retinopathy. Nutrients. 9. 1312. 10.3390/nu9121312.
Reviewer 4 Report
Comments and Suggestions for Authors
The authors have addressed my comments in a constructive and thoughtful manner. The revised manuscript now clearly frames the study as exploratory and hypothesis-generating, which is appropriate given the data limitations. The inclusion of a second cell line (MG63) for experimental validation strengthens the translational relevance. Limitations regarding clinical annotation and follow-up are now explicitly acknowledged, and the discussion has been refined to better contextualize the findings. Overall, the revisions improve the clarity, transparency, and scientific rigor of the work. I support publication in its current form.
Author Response
Comment 1: The authors have addressed my comments in a constructive and thoughtful manner. The revised manuscript now clearly frames the study as exploratory and hypothesis-generating, which is appropriate given the data limitations. The inclusion of a second cell line (MG63) for experimental validation strengthens the translational relevance. Limitations regarding clinical annotation and follow-up are now explicitly acknowledged, and the discussion has been refined to better contextualize the findings. Overall, the revisions improve the clarity, transparency, and scientific rigor of the work. I support publication in its current form.
Response 1: Thank you for your insightful and constructive review. Your suggestions to emphasize the exploratory nature of our study, include MG63 validation, and more transparently acknowledge our clinical limitations have greatly improved our manuscript. We sincerely appreciate your support and believe the paper is better thanks to your thoughtful comments.